# Heterogeneous Folding Intermediates Govern the Conformational Pathway of the RNA Recognition Motif Domain of the Ewing Sarcoma Protein

**DOI:** 10.3390/biom16010033

**Published:** 2025-12-24

**Authors:** Priyanka Kataria, Vishakha Chaudhary, Chandra Bhushan Mishra, Vijay Kumar, Ravi Datta Sharma, Amresh Prakash

**Affiliations:** 1Data Science Laboratory, Amity Institute of Integrative Sciences and Health, Amity University Haryana, Gurugram 122413, Haryana, India; piyadhandhi21@gmail.com (P.K.);; 2Amity Institute of Biotechnology, Amity University Haryana, Gurugram 122413, Haryana, India; 3Department of Pharmacology & Chemical Biology, Baylor College of Medicine, Houston, TX 77030, USA; 4Amity Institute of Biotechnology, Amity University Noida, Noida 201313, Uttar Pradesh, India; 5School of Biotechnology, Shri Mata Vaishno Devi University, Katra 182320, Jammu and Kashmir, India

**Keywords:** RRM, EWS, MD simulation, free-energy landscape, GMM, urea, DMSO

## Abstract

The RNA Recognition Motif (RRM) domain of the Ewing sarcoma (EWS) protein plays a pivotal role in RNA binding and gene regulation, being crucial for its function. However, its structural dynamics are yet to be revealed. Herein, we performed 5.5 μs cumulative molecular dynamics (MD) simulations to investigate the unfolding pathways of the EWS-RRM domain in urea and DMSO across 300–500 K. The unfolding process was characterized by using free-energy landscape (FEL) analysis, hydrogen-bond occupancy, and Gaussian Mixture Model (GMM) clustering. At lower temperatures (300–350 K), the RRM largely retained its native conformation, while extensive unfolding occurred between 400 and 450 K. Results revealed multiple conformational ensembles: native (N), native-like intermediate (I_N_), intermediate (I), and unfolded (U) states, underlying the unfolding pathway of RRM. In urea at 400 K, a long-lived I-state dominated, with transient N and I_N_-populations, whereas in DMSO, the I_N_-state appeared more stable, that transitioned into tightly packed I-states, reflecting a stepwise unfolding via compact intermediates. At 450 K, the protein reached the U-state in both solvents, though unfolding occurred more readily in urea. This study highlights the solvent-dependent unfolding mechanisms and heterogeneous I-states of EWS-RRM, providing insight into its stability, misfolding, and potential relevance to Ewing sarcoma pathogenesis.

## 1. Introduction

EWS (Ewing sarcoma breakpoint region 1/EWS RNA binding protein 1) is a well-known multifunctional protein which plays crucial roles in transcriptional regulation and RNA metabolism and has been implicated in both oncogenesis and neurodegeneration [1,2]. EWS belongs to a FET family of proteins, which includes Fused in Sarcoma (FUS) and TATA-box binding protein Associated Factor 15 (TAF15) [3]. The members of the FET family share high structure and sequence homology (~70%) and are evolutionarily conserved from fish to human [4]. They play diverse roles in physiological cellular functions and are involved in the regulation of RNA transcription and metabolism [5,6]. FET members are DNA/RNA-binding proteins (RBPs) that are composed of an N-terminal serine–tyrosine–glycine–glutamine (SYGQ)-rich low complexity domain (LCD), a central RNA recognition domain (RRM) involved in RNA binding, and a C-terminal zinc finger domain (ZNF) involved in DNA/RNA binding. Additionally, there are several Arg-Gly-Gly (RGG)-rich regions in the C-terminal of FET proteins that influence RNA binding [3,5,7]. The RGG domains are the least conserved regions between members of the FET family. As a result, FET proteins bind nucleic acids via both the RRM and the RGG-ZnF-RGG motifs, which may enable them to bind a wider variety of RNA targets than a single domain. The RGG-ZnF-RGG domain also has a strong affinity for RNA and ssDNA for FUS and EWSR1 [8,9].

Mutations in FET proteins are largely linked to neurodegenerative diseases such as amyotrophic lateral sclerosis (ALS), frontotemporal dementia (FTD), certain sarcomas such as Ewing sarcoma, and other cancers [10,11,12,13]. Ewing sarcoma (EWS) is an aggressive cancer of bone and soft tissue, affecting 1 in 300,000 children in the United States, annually [14]. These tumors are driven by a chromosomal translocation joining the LCD of the EWSR1 with the DNA-binding domain of Friend leukemia integration 1 (FLI1), creating EWS::FLI1, which is a potent fusion oncoprotein responsible for over 85% of EWS tumors and enhances transcription of genes that cause tumorigenesis [15]. For this fusion protein to undergo oncogenic transformation, biomolecular condensation is essential [16,17,18]. This is accomplished through favorable intramolecular and intermolecular interactions within the LCD domains of EWS [15,19]. These interactions facilitate EWS: FLI1 binding to gene enhancers and recruit transcriptional co-factors [20,21,22], thus promoting subsequent oncogenic transactivation.

However, studies have also shown that interdomain interactions between the LCD and RBDs of FUS (and other RBPs) are important for condensate formation [23,24,25]. Furthermore, EWS has an antagonistic effect on the activity of EWS oncofusions, with homotypic LCD-LCD interactions repressing transactivation [26], as well as heterotypic interactions between the RBDs and the LCD of EWS [27,28]. Very recently, Sohn et al. [9] has shown that RBDs of EWS, which include RGG repeat regions and a structured RRM, differentially influence LCD condensate formation in EWS and suggest that electrostatics and polypeptide-chain length likely contribute to this interaction. In addition to the LCD domain, the RRMs also contribute to the self-association/aggregation process, forming fibrillar aggregates in different RBPs. It has been shown that the RRM1 and RRM2 domain of TDP-43 can form aggregates under stress conditions [29,30]. TDP-43 dysfunction and aggregation were also linked through its RRM2 intermediate [31,32]. Similar to TDP-43, FUS-RRM has also been shown to form fibril-like structures [12,33,34,35]. Other studies have shown that RRM domain has an irreversible unfolding and is prone to irreversible self-association [36,37,38,39].

We and others have shown that the folding intermediate in TDP-43 RRMs and FUS RRM is an ensemble of sub-populations of distinct conformations having similar free energy, suggesting the role of conformational heterogeneity in the misfolding and aggregation of these RBPs [35,37,40,41,42,43]. However, very little study of folding/unfolding has been conducted on RRM of EWS protein. We believe that understanding the protein folding pathway and the inherent conformational heterogeneity of the EWS-RRM protein is important to obtain molecular insights into aggregate and condensate formation by EWS protein.

In this study, we systematically investigated the conformational dynamics of the native state of the RRM domain of EWS and examined solvent-induced perturbations on conformational ensembles underlying its unfolding pathway, aiming to elucidate the mechanistic basis of self-association. By perturbing the equilibrium between the native population and excited conformational states through exposure to urea and DMSO at different temperatures, we observed heterogeneous conformational ensembles during unfolding that may enhance the propensity for aggregation and fibril formation. Our analyses indicate that the presence of partially unfolded intermediates, stabilized differently in distinct solvents, facilitates transient interactions that may nucleate biomolecular condensation. These findings highlight the intricate balance between structural stability and dynamic flexibility in the native state, suggesting that even subtle shifts in conformational ensembles can trigger self-association pathways. Collectively, this work underscores the critical role of conformational transitions in protein folding and aggregation, providing mechanistic insights relevant to the molecular basis of pathological fibrillation in EWS and potentially other FET family proteins.

## 2. Materials and Methods

### 2.1. Molecular Dynamics

Multistage molecular dynamics (MDs) simulations of the RRM of EWS were performed in the aqueous solutions of 8 M urea and 8 M DMSO using GROMACS v2024.5 with CHARMm36 force field [44,45,46,47,48]. The initial structural coordinates of the RRM domain (PDB ID: 2CPE) were retrieved from the Protein Data Bank (https://www.rcsb.org/structure/2CPE, 24 February 2023) [49]. Each protein system was solvated in a cubic periodic box with TIP3P water molecules, and the desired cosolvent concentration (8 M urea or 8 M DMSO) was achieved by replacing an appropriate number of water molecules with cosolvent molecules in the ratio of ~7:1 [50,51,52]. Counterions (Na^+^/Cl^−^) were added to neutralize the system [50,53,54]. The structure of the protein in pure water at 300 K was used as a control simulation to characterize the native conformational dynamics of the RRM domain in the absence of cosolvent. To accelerate the protein unfolding in co-solvents, simulations were extended over a temperature range of 300–500 K in incremental stages [50,52,55,56].

Energy minimization was performed in two steps: first using the steepest descent algorithm, followed by the conjugate gradient method, until the maximum force converged below 1000 kJ mol^−1^ nm^−1^. The minimized systems were equilibrated sequentially under NVT conditions for 500 ps, followed by NPT for 1 ns to stabilize temperature, density, and pressure. The V-rescale thermostat and Parrinello–Rahman barostat were employed to maintain a temperature and pressure while under the simulation, respectively [57]. All bond lengths were constrained using the LINCS algorithm, and long-range electrostatic interactions were computed using the Particle Mesh Ewald (PME) method with a 1.0 nm cut-off for both Coulombic and van der Waals interactions. The production simulations were carried out at temperatures ranging from 300 to 500 K in 50 K increments, with each stage simulated for 500 ns, resulting in a cumulative simulation trajectory of 5.5 μs. The integration time step was set to 2 fs, and periodic boundary conditions were applied in all directions. Trajectory analyses were performed using built-in GROMACS tools and in-house Python 3.14.2 scripts [42,58]. The root-mean square deviation (RMSD), root-mean square fluctuation (RMSF), radius of gyration (Rg), and solvent accessible surface area (SASA) were computed to assess structural stability and compactness. The secondary structure evolution was evaluated using DSSP analysis [59].

### 2.2. Fraction of Native Contacts

Using MD Traj, the average change in the fraction of native contacts for the RRM domain in 8 M urea and 8 M DMSO was calculated with respect to simulation time as described by Best et al. [60]. The native contacts, Q (X) in between two amino acid residues i and j of the protein were taken if the distance between a pair of residues (i, j) of 4 or less could be applied to the equation given below.
$$Q(X)=\frac{1}{N}\sum_{i,j}\frac{1}{1+exp[\beta \left(rij\left(X\right)-\lambda r_{ij}^{ο}\right)]}$$
where N is the number of heavy paired atoms (i, j), the paired atoms distance as r_ij_(X) and $r_{ij}^{ο}$ is the distance between atom pairs i and j, in native fold. The smoothing parameter β is taken as 5 Å, and the cut-off for atomic fluctuation factor, λ, is taken as 1.8.

### 2.3. Free-Energy Landscape

The joint probability distribution (P) of the radius of gyration (Rg) and fraction of native contacts was computed and transformed using the Boltzmann relation:
F=−RTln P,
to construct contour plots representing the conformational free-energy landscape (FEL) of the protein [41,61]. These FELs were used to identify major conformational basins and to characterize transitions between folded (N), intermediate (I), and unfolded (U) states, thereby delineating the protein unfolding pathways from the native to denatured conformations.

Additionally, to evaluate the role of hydrogen bonding during solvent-induced unfolding, the occupancy of hydrogen bonds (H-bonds) was analyzed in terms of FELs constructed from the joint probability distribution of H-bonds and native contacts. This approach enabled the visualization of the H-bond stability and disruption contributing to conformational transitions in both urea and DMSO environments during protein unfolding.

### 2.4. Gaussian Mixture Model (GMM)-Based Clustering

A machine learning-based statistical clustering approach, the Gaussian Mixture Model (GMM) [62,63,64], was employed to analyze the MD simulation trajectories and elucidate the structural heterogeneity of the protein conformational ensemble underlying the unfolding pathway. The GMM approach models the overall conformational space as a combination of multiple Gaussian distributions, each representing a distinct conformational basin. Using in-house python script, the MD trajectory data (.xtc and .gro files) were used to extract four fundamental order parameters: RMSD, Rg, SASA, and native contacts (Q) as frames. These features were chosen as collective variables to capture essential aspects of protein during the unfolding simulation. To enable fair comparison across parameters with differing physical units and scales, feature values were standardized using either z-score normalization or by scaling with the interquartile range (IQR), which is defined as
$$IQR=Q_{3}-Q_{1}$$
where $Q_{1}$ is the 25th percentile and $Q_{3}$ is the 75th percentile of the feature distribution. Outliers were identified according to the bounds:
$$\mathrm{Lower}\;\mathrm{bound}=Q_{1}-1.5\times IQR$$
$$\mathrm{Upper}\;\mathrm{bound}=Q_{3}+1.5\times IQR$$

Frames with feature values outside this range were considered outliers and excluded from clustering. After preprocessing, all features were compiled into a feature matrix and clustered using a Gaussian Mixture Model (GMM), whose optimal cluster number was determined by the Bayesian Information Criterion (BIC). Each frame received a probabilistic cluster assignment, delineating metastable ensembles of conformations. Physical meaning was assigned to clusters by computing an unfolding score for each, defined as the mean of *z*-scored RMSD, Rg, SASA, and the negative *z*-scored *Q* values:
$$\mathrm{Unfolding}\;\mathrm{Score}=mean\left({RMSD}_{z},{Rg}_{z},{SASA}_{z},-Q_{z}\right)$$

Clusters were then ranked according to their mean unfolding scores, mapping statistical labels to interpretable native, intermediate, and unfolded protein states.

## 3. Results and Discussion

To investigate the unfolding pathway of the RRM domain of EWS (PDB ID: 2CPE), we carried out multiple MD simulations in the presence of 8 M urea and 8 M DMSO across a temperature range of 300–500 K, each extending up to 500 ns. A control simulation in water at 300 K was also conducted for the comparison. The unfolding dynamics were characterized using key structural parameters (e.g., RMSD, Rg, SASA, and secondary structures), including native contacts, free-energy landscapes, H-bond occupancy, and clustering of ensemble states. A summary of the average changes in the structural parameters is provided in Table 1, and the plots were illustrated in Appendix A. The solvent conditions in combination with increasing temperatures used to accelerate the unfolding process, allowing us to capture different conformational ensembles along the unfolding pathway [65,66,67]. Results suggest that in both solvents (urea and DMSO), the unfolding pathway follows a three-step process involving native (N), intermediate (I), and unfolded (U) states. However, the conformational stabilities of stable and meta-stable intermediate states differed in urea and DMSO. Notably, the I and I_N_-states were more distinctly sampled in DMSO as compared to urea. Together, these results indicate that the RRM domain of EWS undergoes unfolding through three major states (N, I, and U) in both denaturants, but via distinct solvent-dependent mechanisms. Urea promotes a more direct and temperature-enhanced disruption of the native structure, rapidly destabilizing β-sheet elements at the protein core [50,67]. On the other hand, DMSO induces a stepwise unfolding pathway, with clearly distinguishable intermediate (I and I_N_) states, consistent with its preferential action on helical elements and amphiphilic interactions with protein surfaces [50,68]. These findings underscore the differential roles of urea and DMSO in modulating the unfolding landscape of the RRM domain of EWS, highlighting the distinct conformational stabilities and dynamic features that underlie the solvent-dependent unfolding pathways as discussed here in detail.

### 3.1. Structural Dynamics of the RRM Domain of EWS

To examine the structural dynamics of the RRM domain of EWS in urea and DMSO at varying temperatures, the RMSD profiles of the Cα-atoms were analyzed and compared against the reference structure in water at physiological temperature (Appendix A). In water at 300 K, the RRM domain remained highly stable, equilibrating around an average RMSD of 0.11 ± 0.02 nm throughout the simulation. In urea, only marginal perturbations were observed at 300 K and 350 K, with average RMSD values of 0.16 ± 0.28 nm and 0.17 ± 0.53 nm, respectively, indicating preserved conformational dynamics of protein. Similarly, in DMSO, the protein exhibited negligible changes, maintaining an RMSD of 0.12 ± 0.02 nm. Although the structural perturbation initiated in DMSO at 350 K, the system rapidly attains equilibrium around RMSD 0.25 ± 0.04 nm at ~50 ns and is maintained till the end of simulation at 500 ns. In contrast, substantial perturbations emerged at higher temperatures, with three- to five-fold increases in RMSD at 400–450 K, and a dramatic structural disruption occurring within the initial 0–25 ns at 500 K, in both denaturants (urea and DMSO). Thus, we provide a detailed comparative analysis of the structural dynamics in urea and DMSO at 400 K and 450 K, respectively.

Figure 1 shows the probability density distribution plots of the structural parameters, Cα-RMSD, Rg, and SASA for the RRM domain of EWS in urea and DMSO at 400 K. These plots highlight the distinct solvent-dependent effects on the conformational dynamics of the protein. In both denaturants, the RMSD profiles display a bimodal distribution, indicating the coexistence of two distinct conformational populations at 400 K (Figure 1A). In urea, the RMSD distribution is characterized by relatively sharp and narrow peaks, with one population centered at lower RMSD values (~0.20 nm), corresponding to stable conformations, and a second population at higher RMSD values (~0.40 nm), reflecting a partially destabilized state. In contrast, the RMSD distribution in DMSO is broader, with one population confined near RMSD ~0.20 nm and another shifted towards ~1.0 nm, representing a highly deviated and destabilized ensemble. Moreover, the transition in urea appears to occur through a relatively smooth and gradual barrier between the low- and high-RMSD populations. Conversely, in DMSO, the two populations are more distinctly separated, with a pronounced gap between stable and partially unfolded conformations.

The radius of gyration (Rg) is another important structural parameter which defines the structural compactness of a protein [69]. As shown in Figure 1B, the Rg distribution of protein in urea exhibits a single, sharp peak centered around ~1.25 nm, indicating that the protein largely maintains a compact and near-native conformation. Whereas the Rg distribution in DMSO reveals a bimodal distribution, one population is confined near ~1.25 nm, consistent with the compact conformational state, while an additional broader population emerges around ~1.50 nm, reflecting conformations with reduced compactness along the partial unfolding.

Interestingly, we noted that the solvent-accessible surface area (SASA) probability distribution plots of the protein in both urea and DMSO display largely overlapping profiles, with single dominant peaks near ~52–54 nm^2^ (Figure 1C). The SASA provides insights into the degree of solvent exposure and unfolding of the protein. The higher values of SASA typically reflect expanded or partially unfolded conformations with increased exposure of hydrophobic residues to the solvent. Lower SASA values indicate compact, folded states where the hydrophobic core is buried and solvent exposure is minimized. Thus, SASA serves as a sensitive parameter to monitor conformational stability, protein–solvent interactions, and the extent of structural perturbation under different environmental conditions. In urea, the SASA peak is centered around slightly higher values, indicating a modest increase in solvent exposure as compared to DMSO, whereas the SASA distribution in DMSO appears narrower, with its peak shifted marginally lower than urea, indicating less solvent exposed and compact structure of protein.

Notably, at 450 K, the structural parameters probability distribution plots of the RRM domain in urea and DMSO reveal distinct differences in structural stability and unfolding behavior (Figure 2).

Figure 2A shows that the conformational dynamics of protein in both denaturants exhibit multiple peaks in their RMSD distributions which indicates the coexistence of multiple conformational populations at 450 K. In urea, a narrow peak appears around ~0.50 nm, corresponding to stable conformations, along with additional populations emerging at higher RMSD values (~1.75 nm, and ~3.0 nm). This pattern suggests progressive destabilization of the folded structure, with transitions from partially unfolded to fully unfolded states. In contrast, the RMSD distribution in DMSO is broader and more dispersed. While a major population remains centered at ~0.25 nm, with comparatively higher density than in urea, significant fractions shift toward higher RMSD values (~1.0–1.50 nm), with a minor fraction extending further to ~2.0 nm. These observations indicate that both urea and DMSO promote heterogeneous conformational sampling at 450 K, enabling the protein to transition between stable and destabilized states. However, the distinct separation of stable and unfolded states in DMSO indicates a pronounced transition barrier between stable and unfolded populations.

Consistently, the Rg distributions of the protein at 450 K also reveal solvent-dependent differences in structural compactness, with urea and DMSO showing distinct profiles. However, the structural dynamics in both denaturants display a bimodal distribution, reflecting transitions between compact and extended conformations (Figure 2B). In urea, the Rg profile displays sharp peaks, with a minor population centered around ~1.25 nm and a major population corresponding to extended conformations near ~2.0 nm, reflecting progressive destabilization and expended conformations of the protein prolonged up to ~3.0 nm. In contrast, the Rg distribution in DMSO is broader, showing two populations of comparable density centered at ~1.25 nm and ~1.70 nm, separated by a smooth transition barrier. These results highlight the distinct structural compactness and conformational heterogeneity of the protein in urea and DMSO, facilitating transitions between compact and highly extended conformation states.

Furthermore, the SASA probability distributions of the protein in urea and DMSO exhibit overlapping yet distinct patterns, both dominated by a single major peak (Figure 2C). In urea, the distribution is relatively sharp, with the primary population centered around ~55–60 nm^2^ and minor populations appearing in the lower range (~50–54 nm^2^). The SASA distribution in DMSO is broader and slightly shifted toward higher values, with the main population spanning ~55–58 nm^2^ and a smaller fraction near ~50–54 nm^2^. The broader profile in DMSO indicates enhanced conformational heterogeneity and greater solvent penetration as compared to urea, facilitating transitions from compact states to highly solvent-exposed conformations, thereby amplifying unfolding [50,70].

### 3.2. Solvent-Dependent Loss of Native Contacts

Furthermore, to examine the solvent-dependent unfolding dynamics of the RRM domain of EWS, we monitored the fraction of native contacts in the presence of urea and DMSO across different temperatures (300–500 K). Figure 3 shows the time evolution of the fraction of native contacts in urea and DMSO, respectively. As a control, the protein in water at 300 K retained a stable conformation with ~96% of its native contacts (Figure 3A). In the presence of urea at 300 K, unfolding was restricted to a marginal reduction in native contacts, with the protein retaining ~95% of its contacts and thereby preserving its folded conformation.

At 350 K, the unfolding progression became more apparent, as the simulation concluded with a loss of ~8–9% of native contacts. At 400 K, the protein initially exhibited reluctant unfolding during the initial ~50 ns, followed by a gradual loss of ~20% native contacts by ~100 ns. Between ~100 and 200 ns, the unfolding progression appeared to arrest, and an equilibrium was achieved with ~65–70% of native contacts that remain stabilized up to ~425 ns. After that, a further decline of ~10% of native contacts occurred in the last ~75 ns of simulation. At 450 K, a substantial loss of ~50% of native contacts occurred within the initial ~40 ns, accompanied by a transient partial gain in native contacts which can be seen up to ~90 ns. However, this was followed by a progressive loss of native fold, and by ~200 ns, the protein reached to an unfolded population with ~10% of native contacts, which provides elegant evidence of the complete loss of native fold. At 500 K, unfolding was even more drastic, with the entire fold of protein completely losing its structure within the initial ~20–25 ns of the simulation.

Differently, in DMSO at 300 K, the protein retained its native fold with high stability, maintaining ~95% of native contacts throughout the simulation (Figure 3B). Even at 350 K, the fold remained stable, with ~95% of native contacts preserved during the initial ~300 ns. A slight deviation was observed thereafter, leading to a minor loss of ~10% of native contacts, although the overall structure remained intact until the end of the 500 ns trajectory. At 400 K, the native conformation remains largely preserved with ~80–90% of native contacts up to ~150 ns. Followed by a gradual loss of native contacts by ~200 ns, a stable equilibrium was established, having ~50–55% of native contacts that are maintained up to 500 ns. At 450 K, the unfolding pathway could be distinguished into three distinct populations: with an initial rapid loss of ~20% native contacts, followed by the population remains stabilized up to ~50 ns, with ~80% of the native contacts, with the occurrence of sudden structural drift around ~55 ns, resulting in a population with ~50–60% native contacts that observed persistent up to ~150 ns. Finally, there was a gradual transition to an unfolded population during ~150–250 ns. The protein adopted a predominantly unfolded state with ~10% of native contacts that can be observed from ~250 to 500 ns. At 500 K, the entire native fold was rapidly disrupted within the initial ~25–30 ns, and an unfolded state characterized by ~10% of native contacts which dominated the trajectory for the remaining period of the simulation. These results indicate that the RRM domain of EWS follows distinct unfolding mechanisms in the two denaturants, urea and DMSO. At lower temperatures, the protein remains largely preserved in its native fold in both solvents, with substantial loss of native contacts occurring only at elevated temperatures (400–450 K) and the entire native contacts were rapidly lost in few nanoseconds in both denaturants at 500 K.

### 3.3. Time-Resolved Secondary Structure Loss in Urea and DMSO

Figure 4 depicts the time-dependent loss of secondary structure (α-helices and β-sheets) in the RRM domain of EWS in the presence of urea and DMSO. At 400 K, a pronounced loss of α-helices can be seen in DMSO as compared to urea (Figure 4A). Whereas the β-sheet content in urea decreases more rapidly, with a significant loss occurring around ~100 ns. Although the β-structure observed persists up to ~450 ns, it declines sharply during the last 50 ns of simulation. In DMSO, the β-content shows a slight gain in structures during the initial ~0–100 ns, followed by a drop between ~100 and 125 ns (Figure 4B). It remained relatively stable for the next ~75 ns, before it underwent a massive loss of β-structures at ~200 ns.

Surprisingly, the β-contents regained around ~250 ns, and were observed to stabilize up to ~425 ns and even exhibit a modest gain during the last ~75 ns of simulation. At 450 K, the unfolding process accelerates and highlights sharper differences between the two denaturants, urea and DMSO. A massive loss of α-helices is observed in urea compared to DMSO (Figure 4C), underscoring the stronger destabilizing effect of urea on helical structures at elevated temperatures [68,70]. Interestingly, the time evolution plots of secondary structure reveal the formation of non-native helices in DMSO, which may account for the relatively higher helical content observed in this solvent compared to urea (Appendix A). In terms of β-structure, unfolding occurs more rapidly in urea, with almost complete disruption by ~100 ns, indicating a swift collapse of β-sheets. Differently, the β-content in DMSO observed persists for a longer duration, with complete loss occurring only around ~175 ns (Figure 4D), suggesting transient gains and partial stabilizations in β-sheets, during unfolding simulation [42,50].

### 3.4. Free-Energy Landscape of EWS–RRM Unfolding

Another parameter, free-energy landscape (FEL), offers insight into protein structural dynamics, the accessibility of various conformations, native state (N), native-like intermediate (I_N_), intermediate (I), partially unfolded conformation (I_U_), and unfolded (U) states, and the relative thermodynamic stability of states, underlying a protein folding pathway [54,71,72]. The FEL results of RRM of EWS protein in water shows that the conformational dynamics of protein remains well occupied to the native energy basin (N), centered at Rg value ~1.22 nm with ~96–97% native contacts (Appendix A). Figure 5 displays the distribution of conformational states sampled by the protein in urea and DMSO, highlighting the relative stability of the protein and the progression of its unfolding pathway under different denaturing conditions. In urea at 300 K, the conformational dynamics of protein appear reluctant in solvent environment, remaining confined to a single native basin (N) with slight change in Rg ~1.23 nm with ~96% native contacts (Figure 5A). At 350 K, the native basin becomes slightly sallower; however, the conformational population remains overwhelmingly in the N-state (Rg value ~1.24 nm, ~96% native contacts), indicating limited sensitivity to the denaturant (Figure 5B). At 400 K in urea, the protein smoothly escapes the native basin and explores a broader conformational space, shifting toward an I-state (Figure 5C). This state is characterized by an expanded radius of gyration (Rg ~1.25 nm) and a significant reduction in native contacts (~65%), reflecting partial disruption of the native contacts, which results in increased structural heterogeneity.

At 450 K in urea, the N-state population rapidly shifts to the intermediate (I-state) basin (Rg ~1.30 nm; ~60% native contacts) and transiently samples a largely extended conformational (IU*) ensemble (Rg ~1.50 nm; ~30% native contacts) (Figure 5D). The IU* population subsequently progresses toward the unfolded (U) state, spanning a broader conformational space around the Rg ~1.50–2.00 nm with native contacts < ~20%, suggesting the massive loss of the native fold. At 500 K in urea, only the unfolded (U) population is sampled, occupying an even wider conformational space (Rg ~1.50–2.50 nm) and the native contacts reduced to <~10%, signifying the complete loss of the native fold (Figure 5E).

In DMSO at 300 K, the compact and funnel-shaped nature of the FEL implies that the conformational dynamics of protein remain predominantly in an ordered fold. Although it explored broader conformational spaces, its distribution is largely restricted to the N-state around an Rg of ~1.23 nm with 94–96% native contacts (Figure 5F). At 350 K, the N-state population splits into N and I_N_-states, centered at ~95% and ~85% native contacts with Rg values of 1.23 nm and 1.25 nm, respectively (Figure 5G). The smooth transition barrier between the N- and I_N_-states suggests that interconversion between these conformations can occur relatively freely. Figure 5H shows that in DMSO at 400 K, the FEL changes dramatically, exhibiting three distinct minima. The population is confined to a deep energy well around Rg ~1.30 nm with ~80–90% native contacts, indicating N- and I_N_-states. A less-populated basin emerges at Rg ~1.40 nm with ~65% native contacts, representing a transition toward the I-state, as the IU-state ensemble. Furthermore, a broad and shallow energy basin spanning Rg ~1.40–1.60 nm with ~50% native contacts reflect a heterogeneous population within the I-state. At 450 K, the FEL exhibits a rugged topology characterized by multiple shallow basins. A population is observed around Rg ~1.30 nm with ~80% native contacts, corresponding to the I_N_-state. In addition, multiple dispersed populations appear around Rg ~1.40–1.60 nm with ~50–60% native contacts, reflecting transitions into the I-state. Populations centered at larger Rg values (~1.50–1.80 nm) with ~10% native contacts indicate substantial structural disruption and are defined as the U-state (Figure 5I). These results suggest that at 450 K in DMSO, the protein does not occupy a single dominant conformational state, instead sampling a heterogeneous ensemble of partially and fully unfolded conformations, dynamically shifting to I_N_-, I-, and U-states. At 500 K in DMSO, the FEL becomes highly rugged and is dominated by extended conformational states, indicating substantial structural disruption (Figure 5J). The native-like basin is no longer observed, and the conformational population rapidly shifts toward the U-states, characterized by low native contacts (<10–20%) and large Rg values (~1.60–2.00 nm). Considering the appearance and stability of the various conformational ensembles, we performed triplicate simulations in both denaturants at 400 K to validate the results, as shown in Appendix A. The outcomes consistently reproduced the IN- and N-state ensembles, demonstrating that the IN-state exhibits higher conformational stability in DMSO compared to urea.

### 3.5. FEL Mapping of Hydrogen Bonds

In addition, we also constructed the FEL plots using hydrogen bond distribution along with native contacts to examine the unfolding dynamics of the RRM domain of EWS (Figure 6). Hydrogen bonds are crucial for adopting the structural fold and maintaining protein stability, and their disruption is a key mechanism underlying protein unfolding when exposed to denaturants, as urea and DMSO [73].

Figure 6A displays multiple energy minima, reflecting distinct conformational states sampled during the simulations in urea at 400 K. The deep energy basin, located around ~70% native contacts, corresponds to conformations with a higher number of hydrogen bonds and represents relatively stable I-states. A shallow basin appears around ~90% native contacts with fewer hydrogen bonds, indicating partial disruption of the native fold during the transition toward intermediate conformations. The distribution of multiple energy basins suggests that unfolding in urea is initiated by a gradual loss of hydrogen bonds, even when a considerable fraction of native contacts is retained. This result highlights the critical role of urea in destabilizing hydrogen bonds, which triggers structural loosening, followed by a progressive reduction in native contacts as the protein shifts into I-states. Notably, after partial unfolding, the protein occupied a relatively stable I-state, wherein some native hydrogen bonds are regained, defining the appearance of a molten globule (MG)-like state in urea at 400 K. Figure 6B demonstrates that at 450 K in urea, protein unfolding occurs through a broad range of intermediate conformations with varying H-bonds content, accompanied by a gradual loss of native contacts. Unlike the sharp, deep minima corresponding to the I-state population observed at 400 K, the FEL at 450 K is relatively shallow and rugged, with transient I-state populations, before reaching to the fully unfolded conformations with ~10% of native contacts. Further, the H-bonds distribution ~20–40 indicating the unfolded population is largely stabilized with non-native H-bonds.

Differently, the FEL of the protein in DMSO at 400 K exhibits distinct deep and shallow minima, which correspond to the different conformational states of the protein, reflecting the stable states, meta-stable states, and intermediate states (I-states) as a function of hydrogen-bond content and native contacts (Figure 6C). The meta-stable, N, and I_N_-states with partially disrupted H-bonds retains a high percentage of native contacts (85–90%). With further hydrogen-bond disruption, a less populated intermediate state emerges around ~70% native contacts, which subsequently transitions toward the stable I-state, well represented by the deep energy basin at ~50% native contacts. These results suggest that H-bond disruption under the influence of DMSO arises from its aprotic nature, enabling interactions with the protein surface and thereby altering the distribution of native contacts. Interestingly, at 450 K, the FEL also displays three energy basins, ranging from shallow to deep, a sparsely populated shallow basin at ~80% native contacts, another shallow basin corresponding to transient I-states with ~50% of native contacts, and a deep basin associated with fully unfolded state with ~10% native contacts (Figure 6D). Thus, the progressive disruption of H-bonds in DMSO alters the distribution of native contacts, leading to protein unfolding through successive intermediate states, followed by compact and stable I-states, which eventually converge into compact and stable I-states. Collectively, the unfolding of protein in DMSO and urea proceeds through successive intermediate states, maintaining distinct conformational characteristics of I- and I_N_-states, that highlight the critical role of solvent–protein interactions in governing the unfolding pathways of the RRM domain of EWS. Taken together, the free-energy landscapes provide a qualitative view of unfolding pathways of RRM, which reveal multiple I-states with distinct conformational stabilities in urea and DMSO. However, as inverse Boltzmann estimates rely on ergodic sampling, the inferred conformational stabilities underlying the unfolding pathway may be influenced by sampling limitations. Thus, we also performed state populations and occurrence frequencies for transparency by applying GMM.

### 3.6. GMM Clustering of Conformational Landscapes

Further, to identify the heterogeneous conformational states underlying the unfolding pathway of RRM of EWS, a machine learning-based clustering approach, GMM [64,65,66], was applied on the MD trajectories data (Figure 7). The clustering of conformational ensembles of protein in urea at 400 K shows a predominance of I-states, with a significant fraction of N- and I_N_-state. Figure 7A shows that the N-state conformations accounts for ~19.3% of the population, indicating that small proportion of the protein population retains its folded structure. The I_N_-state appeared as a MG-state, characterized by partial unfolded conformation with substantial native-like secondary structure, and contributes ~13.4% to the conformational distribution, whereas the I-state dominates the ensemble with an occupancy of ~67.3%, reflecting the partial unfolding of protein under the influence of urea. The RRM domain of EWS consist of two α-helices (α-helix-1: Leu374-Ala378 and α-helix-2: Pro420-Phe430) and four β-sheets (β1: Ala362-Gln366; β2: Ile398-Ile400; β3: Asp412-Ser416; β4: Lys441-Ser443). The I_N_-state in urea shows a slight gain in non-native helical and β-structure contents. Although the structure of α-helix-2 remains unaltered, the α-helix-1 structure becomes extended Leu374-Gly385. Similarly, during the unfolding, β3 (Ala413-Tyr417) and β4 (Ser438-Ser443) undergo structural extension, accompanied by a marginal loss of β1. The β2 strand melts completely, whereas a new non-native β-sheet (Gly432–Phe435) emerges and remains stable as I-states in urea at 400 K. Thus, this result signifies that although urea destabilizes the native fold, it largely promotes and stabilizes partially unfolded intermediates. Figure 7B reveals three distinct conformational states of the protein in urea at 450 K. The population distribution shows that ~21.0% of the conformational ensembles correspond to relatively less-populated intermediate states.

A larger fraction (~31.1%) represents highly extended structural ensembles, indicating partially unfolded conformations with significant loss of compactness. The dominant cluster, comprising ~47.9% of the population, corresponds to the fully destabilized ensembles of the unfolded state. As the I-state in urea at 450 K, the structural ensemble shows minimal change in the secondary structures of α-helices, with a slight gain in non-native helical content. Whereas, the structurally loosened IU*-state is characterized by a partial loss of native α-helices and β-sheets; however, the structure of new β-strand (Gly432–Phe435) emerged, while the I-state remains consistently populated in the conformational ensemble of the IU*-state. Upon transitioning to the U-state, all secondary-structure elements undergo gradual disruption, leading to a markedly increased population of unfolded conformations enriched with non-native helical structures, indicative of the complete unfolding of the protein. These results provide clear evidence for a three-step unfolding pathway of the RRM domain of EWS, characterized by a substantial shift toward unfolded and extended conformations, while still sampling intermediate states to a lesser extent in urea at 450 K.

In DMSO at 400 K, the GMM clustering analysis reveals that the N/I_N_-state accounts for ~30.7% of the conformational ensemble, indicating that a considerable fraction of the protein retains its folded or native-like structure (Figure 7C). A small fraction (~10.3%) corresponds to loosened intermediate conformations, representing transient structural states along the pathway from the native to intermediate ensemble. With the highest occupancy of ~59.0%, the stable I-state dominates the conformational landscape suggesting that the majority of the protein population exists in partially folded intermediates. The N/I_N_-state ensembles show slight gain in helical (α-helix-1: Leu374-Cys384) as well as β-structures (β2: Ile398-Leu402; β3: Ala413-Tyr417), along with the appearance of a non-native β-structure (Gly432-Phe435). While transitioning to the I-state ensemble, the protein adopts a more loosened conformation, and the β2 strand disappears completely, whereas no significant change is observed in the helical content.

However, in DMSO at 450 K, the protein undergoes significant unfolding, characterized by three distinct conformational populations. As shown in Figure 7D, a minor fraction of structural ensembles (~7.7%) corresponds to the I_N_-state, retaining partially native-like features. A considerable proportion (~43.3%) of the conformations belong to the I-state ensembles, representing partially unfolded conformations with disrupted secondary structure but retaining some compactness, whereas the dominant population (~49.0%) of highly expanded conformations correspond to the U-state population. The I_N_-state in DMSO at 450 K remains relatively less accessible, retaining substantial α-helical and β-sheet secondary structures. As the protein transitions to the I-state, the overall structure becomes more loosened, accompanied by the partial loss of β3 and β4. While β1 and the non-native β-sheet remain intact, the β2 strand is completely lost. The structure as U-state ensemble is dominated by non-native helical structures and the disappearance of native fold, indicating complete unfolding of the protein in DMSO, which proceeds through the I_N_- and I-state intermediates. Considering the appearance and stability of distinct conformational ensembles, we performed triplicate simulations in both denaturants at 400 K, as shown in Appendix A. The results consistently revealed the presence of the I_N_- and N-state ensembles, confirming the greater conformational stability of the I_N_-state in DMSO compared to urea.

Collectively, the results indicate that in urea at 400 K, the protein exhibits a long-lived I-state population, while the N and I_N_ structures appear only transiently. At 450 K, the N- and I_N_-states largely disappear, giving way to the I-state, with unfolding dynamics rapidly progressing from loosely structured conformations to fully unfolded, U-state populations. This behavior aligns with the denaturing mechanism of urea, which interacts directly with the protein backbone and side chains, weakening intramolecular hydrogen bonds and hydrophobic interactions, thereby destabilizing the structure [50,73]. Conversely, the protein follows a markedly different unfolding pathway in DMSO. At 400 K, the populations of N- and I_N_-states are indistinguishable, followed by the emergence of a loosely packed intermediate that subsequently stabilizes into a tightly packed, long-lived I-state. Notably, at 450 K, DMSO promotes a stepwise unfolding process, in which the protein samples distinct intermediate conformations before ultimately reaching the U-state. Furthermore, the core hydrophobic architecture of the RRM domain of EWS is primarily composed of β-sheet segments, which likely explains the more rapid structural loss observed in urea. Urea efficiently disrupts backbone hydrogen bonding and solvates polar groups, preferentially destabilizing β-sheet-rich regions, whereas DMSO predominantly interacts with and destabilizes α-helical elements, accounting for the sequential and stepwise unfolding, where I and I_N_-states are more clearly resolved. Thus, the RRM domain of EWS exhibits distinct conformational dynamics in urea and DMSO, with unfolding proceeding through heterogeneous intermediate states, emphasizing the solvent-specific pathways underlying the protein folding and stability landscape. Thus, the unfolding pathway of the RRM domain of EWS exhibits distinct conformational dynamics in urea and DMSO, proceeding through heterogeneous stable and metastable I-states.

Given its critical role in RNA binding and gene regulation, understanding the stability of the RRM domain can reveal how mutations, fusion events, or environmental stresses compromise its function and contribute to oncogenic processes, such as in Ewing sarcoma. The folding and unfolding dynamics provide insights into structural resilience, conformational plasticity, and the formation of intermediate states, which may be functionally relevant or prone to aggregation. Understanding unfolding pathways of the RRM domain offers a comprehensive view of protein stability, folding mechanisms, misfolding, and solvent-specific effects, informing both fundamental biology and the design of therapeutic strategies aimed at stabilizing or targeting specific conformational states of EWS and protein aggregation mediated disease states.

## 4. Conclusions

In this study, multistage MD simulations were conducted to investigate the unfolding pathway of the RRM domain of EWS in 8 M urea and 8 M DMSO across a temperature range of 300–500 K, respectively. In both denaturants, the conformational dynamics of protein unfolding revealed that it largely retained its native fold at lower temperatures (300–350 K), whereas significant unfolding occurred between 400 and 450 K and the loss of structure observed quite frequently at 500 K. At 400 K, both solvents exhibited bimodal conformational distributions comprising native and partially unfolded populations, which transitioned into a multimodal distribution at 450 K, comprising native, partially unfolded, loosened, and fully unfolded states. FEL and native contact analyses revealed distinct conformational ensembles underlying the unfolding pathway of the RRM domain of EWS. In urea, the N and I_N_ ensembles were transient, while a long-lived I-state was observed at 400 K. In contrast, in DMSO, the I_N_ ensembles were more stable and long-lived at 400 K. At 450 K, the partially unfolded I-states in urea rapidly transitioned into fully unfolded U-state conformations, whereas in DMSO, unfolding followed a stepwise mechanism with well-resolved N-, I_N_-, and I-states before reaching the U-state. GMM clustering corroborated these observations, showing that at moderate temperatures, partially folded intermediates dominated in both solvents. However, at higher temperatures, urea promoted extended U-state conformations, while DMSO stabilized sequential I-state populations. In urea at 400 K, the conformational populations were distributed as N (19.3%), I_N_ (13.4%), and I-states (67.3%), which shifted at 450 K toward I (21%), IU* (31.1%), and U (47.9%) populations. These results suggest that urea preferentially disrupts β-sheet-rich regions by solvating polar groups and weakening backbone hydrogen bonds, consistent with hydrogen-bond distribution patterns. In DMSO, the populations at 400 K included N/I_N_ (30.7%), IU (10.3%), and stable I-states (59%), while at 450 K, I_N_- population (7.7%), I- (43.3%) and U-populations (49%) were observed. Mechanistically, DMSO perturbs α-helical elements, thereby stabilizing intermediate conformations. Taken together, the unfolding of the EWS–RRM domain proceeds through heterogeneous, stable, and metastable I-states with distinct solvent-dependent dynamics. Thus, considering the critical role of the RRM in RNA binding and oncogenesis, these observations provide valuable insights into its conformational resilience and potential therapeutic avenues for EWS-associated pathologies.

## Figures and Tables

**Figure 1 biomolecules-16-00033-f001:**
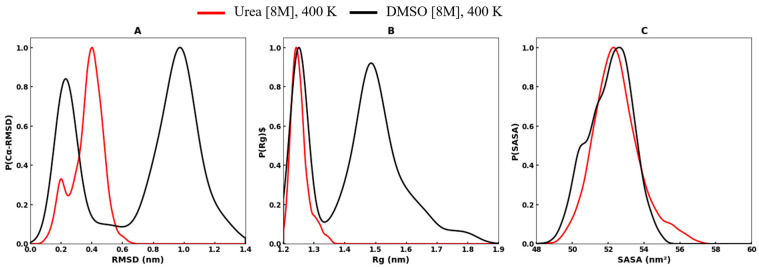
The structural parameters, probability distribution plots (**A**) Cα- RMSD, (**B**) radius of gyration (Rg), and (**C**) SASA of protein, and RRM domain of EWS (PDB ID: 2CPE) during the simulation in urea and DMSO at 400 K. The conformational profile in urea is shown in red, while the system in DMSO is shown in black, illustrating solvent-dependent differences in structural dynamics.

**Figure 2 biomolecules-16-00033-f002:**
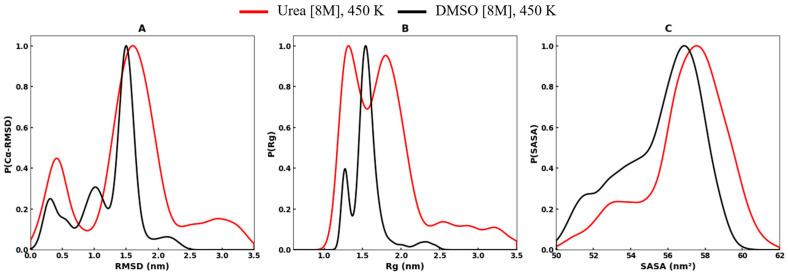
The probability distribution plots of RRM domain of EWS in urea and DMSO at 450 K for (**A**) Cα- RMSD, (**B**) radius of gyration (Rg), and (**C**) SASA of protein. The plot for protein in urea is shown in red color, whereas in DMSO, it is shown in black color.

**Figure 3 biomolecules-16-00033-f003:**
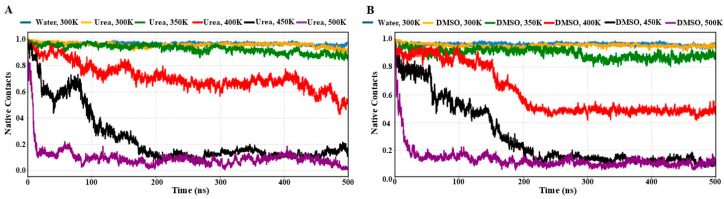
The time evolution plots of the native contacts of RRM domain of EWS during 500 ns simulations at different temperatures. (**A**) in 8 M urea and (**B**) in 8 M DMSO. For comparison, the native contacts of protein in pure water are shown in sky blue in both plots.

**Figure 4 biomolecules-16-00033-f004:**
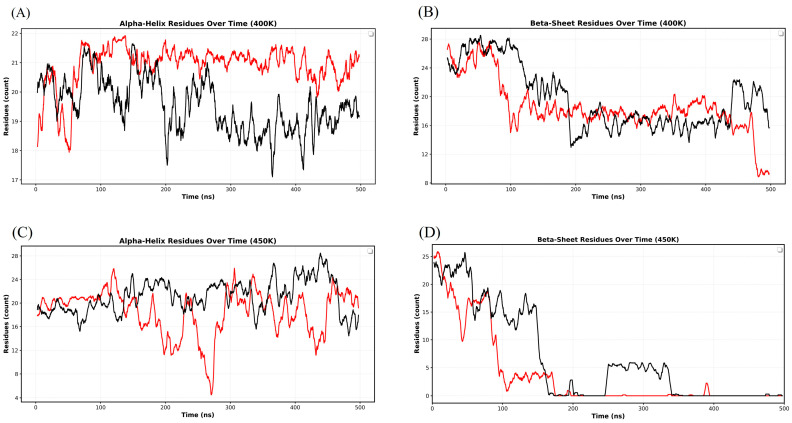
The loss of secondary structure (α-helices and β-sheets) contents of RRM domain in urea and DMSO, respectively. (**A**) the time-dependent loss of α-helices structures in urea and DMSO at 400 K, (**B**) the evolution of the loss of β-sheets in urea and DMSO at 400 K, (**C**) the loss of α-helices in urea and DMSO at 450 K, and (**D**) the loss of β-structures in urea and DMSO at 450 K. The plot of the secondary structure in urea is shown in red color and plots in DMSO are shown in black color.

**Figure 5 biomolecules-16-00033-f005:**
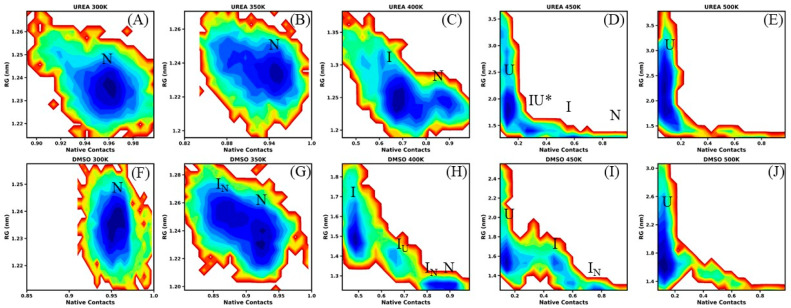
The conformational free-energy landscape (FEL) of protein, and RRM of EWS constructed using the joint probability distribution of radius of gyration (Rg) and the native contacts, across the temperatures 300–500 K (**A**–**E**) in urea 8 M and (**F**–**J**) in DMSO 8 M. The contour plots depict the major conformational basins corresponding to the native (N), native-like intermediate (I_N_), intermediate (I), in transition intermediate (IU), largely extended conformational ensemble (I_U_), and unfolded (U) states of the protein. The depth and spread of the energy minima represent the stability and heterogeneity of conformational ensembles in 8 M urea and 8 M DMSO, respectively. The color is scaled according to kcal mol^−1^. Key minima (basins) in the maps are marked. The surrounding contour layers with gradually increasing energy (deep blue → red) indicate lower- to → higher-energy conformational ensembles.

**Figure 6 biomolecules-16-00033-f006:**
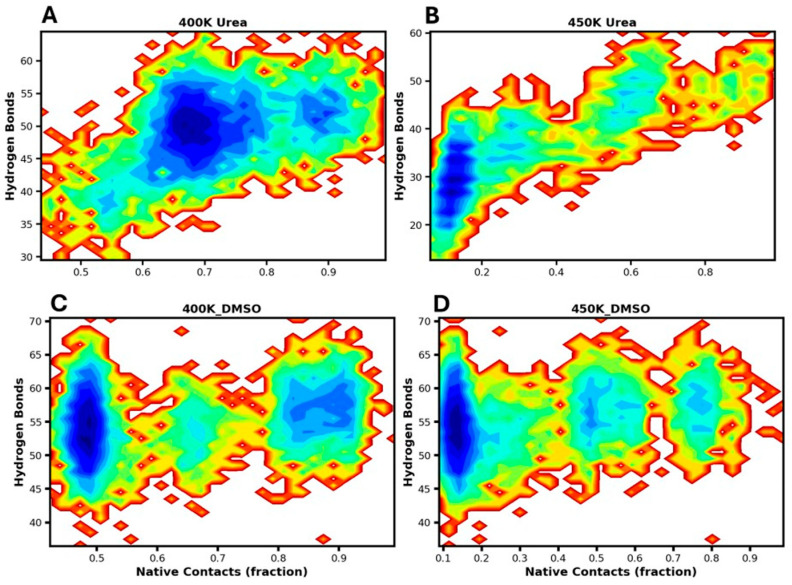
Free-energy landscape of the protein in urea (**A**,**B**) and DMSO (**C**,**D**) as a function of hydrogen-bond content and native contacts at 400 K and 450 K, respectively. Distinct deep minima correspond to stable conformations with high native contacts and preserved hydrogen bonds, while intermediate states with partially disrupted hydrogen bonds are sparsely populated, and the unfolded population with native contacts ≤20% are largely stabilized non-native H-bonds.

**Figure 7 biomolecules-16-00033-f007:**
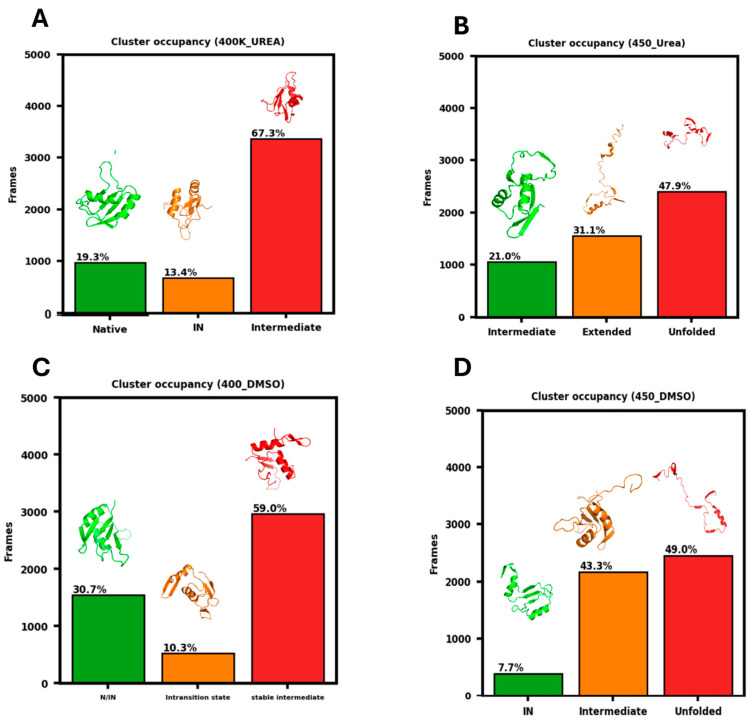
Gaussian Mixture Model (GMM)-based clustering analysis of the MD trajectories of the RRM domain of EWS, depicting heterogeneous conformational ensembles underlying the unfolding pathway in 8 M urea (**A**,**B**) and 8 M DMSO (**C**,**D**) at 400 K and 450 K, respectively. The structural snapshots of representative conformational ensembles as shown corresponding to clusters.

**Table 1 biomolecules-16-00033-t001:** Average value of the structural parameters of RRM of EWS, during the simulations in urea and DMSO, at different temperatures, respectively.

Temp.	RMSD (nm)	Rg (nm)	Native Contacts (%)	SASA (nm^2^)
**RRM of EWS, Water**				
300 K	0.11 ± 0.02	1.22 ± 0.01	96.01%	50.47 ± 0.59
**RRM of EWS, Urea [8 M]**				
300 K	0.16 ± 0.28	1.23 ± 0.01	95.6%	50.28 ± 0.66
350 K	0.17 ± 0.53	1.23 ± 0.01	92.66%	50.73 ± 0.88
400 K	0.37 ± 0.99	1.25 ± 0.03	71.3%	52.48 ± 1.31
450 K	1.55 ± 0.73	1.80 ± 0.52	25.14%	56.92 ± 2.18
500 K	2.02 ± 0.47	2.18 ± 0.44	09.85%	58.99 ± 1.67
**RRM of EWS, DMSO [8 M]**				
300 K	0.12 ± 0.02	1.23 ± 0.01	95.3%	50.22 ± 0.57
350 K	0.25 ± 0.04	1.24 ± 0.01	89.69%	51.05 ± 0.64
400 K	0.72 ± 0.36	1.43 ± 0.15	61.2%	52.10 ± 1.20
450 K	1.22 ± 0.48	1.55 ± 0.21	29.6%	55.56 ± 2.15
500 K	1.65 ± 0.34	1.82 ± 0.30	13.42%	56.90± 1.36

## Data Availability

The original contributions presented in this study are included in the article/Appendix A. Further inquiries can be directed to the corresponding author.

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
