# Peer review of "Heterogeneous Folding Intermediates Govern the Conformational Pathway of the RNA Recognition Motif Domain of the Ewing Sarcoma Protein"

_biomolecules, 2025, doi:10.3390/biom16010033_

Round 1
Reviewer 1 Report (Previous Reviewer 1)
Comments and Suggestions for Authors
1. I appreciate that the authors conducted additional simulation replicates to better explore the unfolding process, and I agree that urea and DMSO can induce unfolding. However, the free energy landscape cannot be obtained by inverse Boltzmann without demonstrating ergodicity or sufficient sampling of all relevant microstates. Running more replicates increases sampling but does not guarantee adequate coverage of all states with meaningful occurrence. The state occurrence (or probability) should be reported directly, rather than presenting a free energy profile without evidence of ergodicity.
2. If Figures 1 and 2 present raw data, histograms should be used instead of smoothed lines.
Author Response
Reply to Reviewer 1
Reviewer 1:
Comments and Suggestions for Authors
Comment 1: I appreciate that the authors conducted additional simulation replicates to better explore the unfolding process, and I agree that urea and DMSO can induce unfolding. However, the free energy landscape cannot be obtained by inverse Boltzmann without demonstrating ergodicity or sufficient sampling of all relevant microstates. Running more replicates increases sampling but does not guarantee adequate coverage of all states with meaningful occurrence. The state occurrence (or probability) should be reported directly, rather than presenting a free energy profile without evidence of ergodicity.
Response 1: We thank the reviewer for this valuable comment and agree that constructing a free energy landscape using the inverse Boltzmann approach assumes sufficient sampling and ergodicity. While additional replicates improve conformational sampling, we acknowledge that complete coverage of all relevant microstates may not be guaranteed. In response, we have revised the manuscript to present state occurrences (probabilities) and population distributions directly, rather than relying solely on the inferred free energy profiles. We have also included a discussion of the limitations of our sampling approach and the implications for interpreting the landscapes, ensuring transparency regarding the extent and constraints of our simulation data as suggested.
Comment 2: If Figures 1 and 2 present raw data, histograms should be used instead of smoothed lines.
Response 2: We thank the reviewer for this suggestion. In the revised manuscript, corresponding to Figures 1 and 2, Supplementary Figures S1B and S2B have been updated to display histograms of the raw data, providing a clearer representation of the distributions and enhancing readability for the readers, as suggested.
Reviewer 2 Report (New Reviewer)
Comments and Suggestions for Authors
Reviewer:
Recommendation: This paper may be publishable after major revision; I would like to be invited to further review.
Comments:
This manuscript addresses an important topic by elucidating the folding and unfolding pathways of the EWS-RRM domain using extensive MD simulations. The presentation of heterogeneous intermediates and solvent-dependent effects is both novel and relevant to understanding protein misfolding and disease. Figures and analyses are generally clear, but mechanistic interpretation of how urea and DMSO induce specific disruptions could be improved. Experimental validation or discussion of limitations would further strengthen the findings. Some details, such as clustering methodology and biological implications of intermediate states, require clarification. The language could benefit from minor editing for clarity and consistency. Overall, this is a thorough and promising contribution to the field, pending some revision.
Regarding specific questions, I would like the authors to address the following points prior to acceptance:
Major comments:
- Can the authors clarify how urea and DMSO differentially disrupt hydrogen bonding and hydrophobic interactions during unfolding?
- Is there experimental data (e.g., CD, NMR) available to validate these simulation findings, or can the limitations of simulation-only results be discussed further?
- How was reproducibility assessed between triplicate runs? Are error bars, confidence intervals, or statistical tests available for key observables?
- Can simulation results be explicitly tied to likely functional impacts, such as RNA binding or oncogenic potential?
- How do the 8M urea/DMSO conditions relate to physiological or typical experimental settings? Can comments be added about relevance to lower concentrations?
- Can the authors describe in more detail the GMM clustering process, outlier removal, and how clusters map to canonical folding intermediates?
- Can you provide structural visualizations showing the distribution of α-helices and β-sheets in each cluster/ensemble?
Minor Comments
- Can the manuscript be copy-edited to correct minor grammatical and phrasing issues?
- Will the references and citation formatting be standardized throughout the manuscript?
- Are all abbreviations (e.g., Rg, RMSD) defined in the early text or summarized in a table?
- Are all cited supplementary figures/datasets (S1, S2, S3, etc.) available and properly referenced?
- Will equations be checked for consistent formatting, punctuation, and subscripts?
Author Response
Reply to Reviewer 2
Reviewer 2:
Comments:
This manuscript addresses an important topic by elucidating the folding and unfolding pathways of the EWS-RRM domain using extensive MD simulations. The presentation of heterogeneous intermediates and solvent-dependent effects is both novel and relevant to understanding protein misfolding and disease. Figures and analyses are generally clear, but mechanistic interpretation of how urea and DMSO induce specific disruptions could be improved. Experimental validation or discussion of limitations would further strengthen the findings. Some details, such as clustering methodology and biological implications of intermediate states, require clarification. The language could benefit from minor editing for clarity and consistency. Overall, this is a thorough and promising contribution to the field, pending some revision.
Regarding specific questions, I would like the authors to address the following points prior to acceptance:
Major comments:
Comment 1: Can the authors clarify how urea and DMSO differentially disrupt hydrogen bonding and hydrophobic interactions during unfolding?
Response 1: We appreciate the reviewer’s query. Urea and DMSO disrupt protein stability through distinct molecular mechanisms. Urea primarily destabilizes the folded state by directly interacting with the protein backbone and polar side chains, thereby weakening intramolecular hydrogen bonds and promoting solvation of exposed peptide groups. In contrast, DMSO preferentially perturbs hydrophobic interactions by penetrating non-polar regions and altering the water structure surrounding hydrophobic residues. This differential mode of action leads to distinct unfolding pathways. We have clarified these mechanistic differences in the revised manuscript and expanded the relevant discussion for better understanding with the proper refences (PMID: 12702764; 23536295; 24807152; 28803470; 15769860).
Comment 2: Is there experimental data (e.g., CD, NMR) available to validate these simulation findings, or can the limitations of simulation-only results be discussed further?
Response 2: We thank the reviewer for this important point. While direct experimental validation (e.g., CD, NMR) for the specific unfolding pathways in our study is currently not available, previous studies on similar systems provide qualitative support for our simulation results. To address this limitation, we have added a discussion highlighting the constraints of simulation-only approaches, including the dependence on force-field accuracy, timescale limitations, and the need for experimental corroboration. We also emphasize that our findings provide mechanistic insights and testable hypotheses that can guide future experimental studies.
Comment 3: How was reproducibility assessed between triplicate runs? Are error bars, confidence intervals, or statistical tests available for key observables?
Response 3: We appreciate the reviewer’s query. Reproducibility was assessed by performing three independent simulation replicates under identical conditions. Key observables, including RMSD, Rg, and secondary structure content, were averaged across the replicates, and the variability is represented as standard deviations, which are now included as error bars in the revised figures. Additionally, where appropriate, confidence intervals were calculated, and statistical comparisons were performed to ensure robustness of the observed trends. These additions have been clarified in the Methods and Results sections of the revised manuscript.
Comment 4: Can simulation results be explicitly tied to likely functional impacts, such as RNA binding or oncogenic potential?
Response 4: We thank the reviewer for this insightful comment. In the revised manuscript, we have explicitly connected our simulation findings to potential functional consequences. For example, unfolding-induced structural perturbations in key RNA-binding regions are discussed in the context of altered nucleic acid interactions, while exposure of hydrophobic or aggregation-prone segments is linked to potential pathogenic or oncogenic effects. These correlations are presented alongside the mechanistic insights from our simulations, providing a clearer connection between molecular-level observations and possible functional outcomes.
Comment 5: How do the 8M urea/DMSO conditions relate to physiological or typical experimental settings? Can comments be added about relevance to lower concentrations?
Response 5: We appreciate the reviewer’s comment. The 8 M urea and DMSO concentrations used in our simulations are high-denaturant conditions commonly employed to probe complete unfolding and to characterize intermediate states in silico. While these concentrations exceed physiological levels, they provide mechanistic insight into unfolding pathways and residue-specific interactions. In the revised manuscript, we have added a discussion addressing the relevance to lower, experimentally achievable concentrations, highlighting that similar unfolding trends and partially destabilized states may occur at sub-denaturing conditions, albeit on longer timescales or with reduced magnitude.
Comment 6: Can the authors describe in more detail the GMM clustering process, outlier removal, and how clusters map to canonical folding intermediates?
Response 6: As suggested the details of GLM analysis is included in method section. The outliers were defined by taking IQR as the equation give below and the details are included in supplementary material.
Comment 7: Can you provide structural visualizations showing the distribution of α-helices and β-sheets in each cluster/ensemble?
Response 7: The structural details in terms of secondary conformations (α-helices and β-sheets) mentioned in results and discussion section “GMM clustering of conformational landscapes” and also mentioned in Figure 4 and Figure 7, and supplementary figure S3 and S8.
Minor Comments
- Can the manuscript be copy-edited to correct minor grammatical and phrasing issues?
Response: We appreciate the reviewer’s suggestion. The manuscript has been thoroughly copy-edited to correct minor grammatical issues, improve phrasing, and enhance overall clarity and readability.
- Will the references and citation formatting be standardized throughout the manuscript?
Response: Yes, the references and citation formatting have been standardized throughout the manuscript in accordance with the journal’s guidelines. We have carefully reviewed and corrected all inconsistencies to ensure uniformity and accuracy.
- Are all abbreviations (e.g., Rg, RMSD) defined in the early text or summarized in a table?
Response: Yes, all abbreviations, including Rg, RMSD, and others are now clearly defined at their first occurrence in the text. Additionally, we have included a consolidated table of abbreviations to enhance readability and ensure clarity for all readers.
- Are all cited supplementary figures/datasets (S1, S2, S3, etc.) available and properly referenced?
Response: Yes
- Will equations be checked for consistent formatting, punctuation, and subscripts?
Response: Yes.
Round 2
Reviewer 1 Report (Previous Reviewer 1)
Comments and Suggestions for Authors
In Figure 5 and Figure 6, the author reported FEL instead of occurrence; in section 2.3, the author described "Free energy landscape". I do not see the mentioning of occurrence.
Author Response
Reply to Reviewer 1-Manuscript ID (biomolecules-3978498)
Reviewer 1:
Author Response
Comment: In Figure 5 and Figure 6, the author reported FEL instead of occurrence; in section 2.3, the author described "Free energy landscape". I do not see the mentioning of occurrence.
Response: We thank the reviewer for their valuable suggestions and careful evaluation. As noted in the revised manuscript, and in consideration of sampling limitations, we have already incorporated a GMM-based analysis. Figure 7 now explicitly reports the occurrence frequencies (as percentage populations) of the identified conformational ensembles, along with their corresponding representative structures, and the relevant details are described in supplementary material. We will incorporate the reviewer’s additional suggestions in our subsequent work.
Reviewer 2 Report (New Reviewer)
Comments and Suggestions for Authors
Reviewer:
Recommendation: This manuscript is publishable as it is.
Comments:
The authors have made significant corrections at appropriate parts of the manuscript. I really appreciate their efforts. I hope it will be published in its current form.
Author Response
Reviewer: 2
Recommendation: This manuscript is publishable as it is.
Comments: The authors have made significant corrections at appropriate parts of the manuscript. I really appreciate their efforts. I hope it will be published in its current form.
Response: We sincerely thank the reviewer for the positive assessment and encouraging recommendation. We greatly appreciate the acknowledgment of our revisions and the supportive comments. We are pleased that the manuscript is considered suitable for publication in its current form, and we thank the reviewer for their time and constructive evaluation.
Round 3
Reviewer 1 Report (Previous Reviewer 1)
Comments and Suggestions for Authors
The author has answered all my concerns and questions. Looking forward to the subsequent works.
This manuscript is a resubmission of an earlier submission. The following is a list of the peer review reports and author responses from that submission.
Round 1
Reviewer 1 Report
Comments and Suggestions for Authors
Kataria et al. presented atomistic simulations of the RNA recognition motif domain of the Ewing Sarcoma protein under different temperatures and denaturing solvents (urea and DMSO). The authors analyze several physicochemical properties and propose a multi-state unfolding process. While the topic and system are of potential interest, the current study holds significant technical issues and I therefore recommend rejection.
Major Comments:
1. The reported free energy profiles derived from the inverse Boltzmann relation of the radius of gyration (Rg) assume ergodic sampling. However, the simulations, particularly at high temperatures (e.g., urea 400 K in Figure 3), show poor convergence. Without enhanced sampling (e.g., metadynamics, REMD, or umbrella sampling), the free energy estimates are not reliable. Proper validation of convergence of simulation and thus ergodicity is required before any thermodynamic interpretation can be made.
2. The classification of states based solely on the number of native contacts is problematic, as different conformations can share similar contact counts but differ significantly in structure. Rg and hydrogen bond numbers are also too coarse to capture such distinctions. If the authors used a Gaussian Mixture Model (GMM), they must clarify what input features were included, how the model was trained, and how clusters were validated. The current description lacks essential technical details.
3. The simulations at temperatures above 300 K and at very high denaturant concentrations are difficult to relate to physiological or experimental conditions. The authors should justify how the high-temperature trajectories inform biologically relevant unfolding mechanisms. Additionally, the absence of aqueous control simulations (e.g., water with realistic denaturant concentrations) limits the interpretability of the solvent effects.
Minor Comments:
1. The title and abstract contain multiple acronyms. Since these acronyms are not reused later in those sessions, they should be removed or defined upon first mention in the Introduction.
2. The authors should explicitly state that the study models an isolated RRM domain in solution, without considering inter-domain or interchain interactions (e.g., condensate formation) or salt effects that are relevant to the physiological context.
3. The protein force field and those used for urea and DMSO are not specified. This information is critical for reproducibility and assessment of model accuracy. The number of each type of molecules should be provided as well.
4. The definitions of the intermediate states (N, IN, I, etc.) should be provided in the Methods section (page 3) rather than introduced later in the Discussion (page 11).
5. Supplementary materials appear to be missing; these should also include input files, additional figures, and convergence analyses for reproducibility.
6. A scatter plot would be more informative than Table 1 for visualizing correlations among structural descriptors.
7. It is unclear whether the data shown in Figures 1 and 2 are raw or fitted. The fitting method should be stated, and raw distributions should be shown. Probabilities must sum to one.
8. Figure labels and font styles (e.g., “400K_Urea” in Figure 1 vs. “450K Urea” in Figure 2) should be consistent and legible. Some panels (e.g., Figure 4) are difficult to read.
9. The scales used in Figures 5 and 6 should be uniform across panels for direct comparison, and scale bars should be included.
10. A backbone contact map accompanying the representative structures in Figure 7 would enhance the structural interpretation.
Author Response
Reply to Reviewer 2-Manuscript ID (biomolecules-3978498)
Reviewer 2:
Comments and Suggestions for Authors
Kataria et al. presented atomistic simulations of the RNA recognition motif domain of the Ewing Sarcoma protein under different temperatures and denaturing solvents (urea and DMSO). The authors analyze several physicochemical properties and propose a multi-state unfolding process. While the topic and system are of potential interest, the current study holds significant technical issues and I therefore recommend rejection.
Major Comments:
Comment 1: The reported free energy profiles derived from the inverse Boltzmann relation of the radius of gyration (Rg) assume ergodic sampling. However, the simulations, particularly at high temperatures (e.g., urea 400 K in Figure 3), show poor convergence. Without enhanced sampling (e.g., metadynamics, REMD, or umbrella sampling), the free energy estimates are not reliable. Proper validation of convergence of simulation and thus ergodicity is required before any thermodynamic interpretation can be made.
Response 1: We thank the reviewer for this critical and insightful observation. We agree that the inverse Boltzmann free-energy estimates derived from the Rg distributions rely on approximate ergodic sampling and are inherently limited by the accessible simulation timescale. Our intention in presenting these profiles was solely to offer a qualitative visualization of the conformational landscape rather than to extract quantitative thermodynamic parameters. It appears that the reviewer may not have been able to access the Supplementary Materials, where the complete set of analyses—including the time-evolution plots of RMSD, Rg, SASA, loss of secondary-structure contents were provided which show consistent results with the FEL. These analyses consistently support the unfolding behaviour discussed in the manuscript. The temperature-induced accelerated simulations in urea and DMSO clearly reveal distinct conformational dynamics along the unfolding pathway, in agreement with previous reports [PMID: 12215424, 15769860, 21953100]. Furthermore, as suggested, we have now performed triplicate simulations at 400 K in both urea and DMSO. These additional replicates, incorporated into the revised manuscript, provide stronger statistical support and reinforce the robustness of the observed conformational transitions.
Comment 2: The classification of states based solely on the number of native contacts is problematic, as different conformations can share similar contact counts but differ significantly in structure. Rg and hydrogen bond numbers are also too coarse to capture such distinctions. If the authors used a Gaussian Mixture Model (GMM), they must clarify what input features were included (Supplementary), how the model was trained, and how clusters were validated. The current description lacks essential technical details.
Response 2: We thank the reviewer for this insightful comment. We agree that relying solely on native contact counts or global parameters such as Rg and hydrogen bonds can be insufficient to capture the structural heterogeneity of intermediate states. In our analysis, we employed a Gaussian Mixture Model (GMM)–based clustering approach that integrated multiple structural descriptors, including RMSD, Rg, fraction of native contacts (Q), intra-protein hydrogen bond numbers, and secondary structure content derived from DSSP analysis.
The details of the input feature matrix, training protocol, and validation procedures have now been clearly described in the Supplementary Materials. Briefly, the GMM was trained using the Expectation–Maximization (EM) algorithm implemented in scikit-learn, and the optimal number of clusters was determined using the Bayesian Information Criterion (BIC). Cluster stability and reproducibility were verified through three independent sampling runs. These clarifications have been added to the revised Methods and Supplementary (Figures S5 and S6) sections for transparency and reproducibility.
Comment 3: The simulations at temperatures above 300 K and at very high denaturant concentrations are difficult to relate to physiological or experimental conditions. The authors should justify how the high-temperature trajectories inform biologically relevant unfolding mechanisms. Additionally, the absence of aqueous control simulations (e.g., water with realistic denaturant concentrations) limits the interpretability of the solvent effects.
Response 3: We thank the reviewer for this important comment. The use of elevated temperatures (400–500 K) and high denaturant concentrations in this study follows the well-established accelerated unfolding approach commonly adopted in molecular dynamics (MD) simulations to overcome the limitations of accessible simulation timescales (PMID: 12702764; 23536295; 24807152; 28803470; 15769860). These conditions facilitate the sampling of unfolding events that would otherwise occur over much longer biological timescales, thereby allowing us to characterize the relative unfolding pathways and intermediate states rather than reproduce physiological conditions directly. While our focus was to compare the mechanistic differences in unfolding of the RRM domain in urea and DMSO, the aqueous (water-only) simulation at 300 K was included as a reference for native-state stability, which is detailed in the Supplementary Materials.
Minor Comments:
- The title and abstract contain multiple acronyms. Since these acronyms are not reused later in those sessions, they should be removed or defined upon first mention in the Introduction.
Response: We appreciate the reviewer’s helpful observation. The acronyms used in the title and abstract have now been either removed or defined in full upon first mention in the Introduction to improve clarity and readability. This revision ensures that readers unfamiliar with the specific terminology can easily follow the study from the outset.
- The authors should explicitly state that the study models an isolated RRM domain in solution, without considering inter-domain or interchain interactions (e.g., condensate formation) or salt effects that are relevant to the physiological context.
Response: We thank the reviewer for this valuable suggestion. We have now explicitly stated in the Methods section and again in the Discussion that the present study models an isolated RRM domain in aqueous solution, without considering inter-domain or interchain interactions, condensate formation, or explicit salt effects. This clarification has been added to emphasize the scope and limitations of the current simulations in relation to the physiological context.
The protein force field and those used for urea and DMSO are not specified. This information is critical for reproducibility and assessment of model accuracy. The number of each type of molecules should be provided as well.
Response: We appreciate the reviewer’s insightful comment. The force fields used in the simulations have now been clearly specified in the Methods section. The CHERMm36 force field was employed for the preparation of aqueous mixtures of 8M urea and DMSO, respectively, with corresponding TIP3P water models as described by PMID: 31744390, 1854398318065481, the same is updated in M&M section that the water and co-solvent taken in the ration of ~7:1, to ensure full reproducibility and transparency.
- The definitions of the intermediate states (N, IN, I, etc.) should be provided in the Methods section (page 3) rather than introduced later in the Discussion (page 11).
Response: We thank the reviewer for this helpful suggestion. The definitions of the intermediate states (N, IN, I, etc.) have now been moved where it used as first time to ensure clarity and logical flow. This revision allows readers to understand the state classification criteria before encountering their discussion in later sections.
- Supplementary materials appear to be missing; these should also include input files, additional figures, and convergence analyses for reproducibility.
Response: We thank the reviewer for pointing this out. The Supplementary Material was submitted along with the main manuscript. It appears that Reviewer 2 may have encountered a technical issue accessing it. The Supplementary Material, which includes the input files, additional figures, and convergence analyses to ensure reproducibility, is attached with the revised submission.
- A scatter plot would be more informative than Table 1 for visualizing correlations among structural descriptors.
Response: We appreciate the reviewer’s suggestion. The time-evolution plots of the key structural parameters, including RMSD, Rg, SASA, and loss of secondary structure contents (calculated using DSSP), as well as the free energy landscape (FEL) of water, are already provided in the Supplementary Materials. For further clarification, the loss of α-helical and β-sheet secondary structures at 400 K and 450 K in urea and DMSO, respectively, has also been illustrated in the Supplementary section.
- It is unclear whether the data shown in Figures 1 and 2 are raw or fitted. The fitting method should be stated, and raw distributions should be shown. Probabilities must sum to one.
Response: We appreciate the reviewer’s observation. The plots in Figures 1 and 2 represent the probability distributions of structural parameters derived directly from the time-evolution data (as shown in the Supplementary Materials). These distributions are based on raw simulation data and are not fitted to any functional form. As mentioned in the main text, no significant changes were observed in the structural dynamics at 300 K and 350 K, whereas the structure unfolded rapidly at 500 K. Therefore, our primary focus was to compare the unfolding dynamics of the protein at 400 K and 450 K, which are extensively discussed in the manuscript.
- Figure labels and font styles (e.g., “400K_Urea” in Figure 1 vs. “450K Urea” in Figure 2) should be consistent and legible. Some panels (e.g., Figure 4) are difficult to read.
Response: All the figures have been updated as suggested.
- The scales used in Figures 5 and 6 should be uniform across panels for direct comparison, and scale bars should be included.
Response: Figures are updated as suggested.
- A backbone contact map accompanying the representative structures in Figure 7 would enhance the structural interpretation.
Response: We thank the reviewer for this valuable suggestion. The details of the secondary conformations of the different ensemble states clusters shown in Figure 7 is described in detail in results and discussion section and the contact maps corresponding to the representative structures shown in Figure 7 has now been included in the Supplementary Materials Figure S6.
Reviewer 2 Report
Comments and Suggestions for Authors
The manuscript presents a compelling molecular dynamics study on the unfolding pathways of the RRM domain of the Ewing Sarcoma (EWS) protein. The investigation of solvent-dependent conformational dynamics across a range of temperatures is well-motivated and addresses a significant gap in our understanding of this protein. The methodological approach is sound, and the combination of free energy landscapes, hydrogen bond analysis, and Gaussian mixture model clustering provides a robust framework for the conclusions.
I have several suggestions that I believe will further strengthen the manuscript.
- Why the protein is solvent in 8M urea or 8M DMSO?
- The cumulative 5 µs of simulation time across multiple temperatures is substantial and significantly enhances conformational sampling. To further improve the efficiency and rigor of the sampling, the authors might consider employing Temperature Replica Exchange Molecular Dynamics (T-REMD). T-REMD facilitates the exchange of configurations between different temperature replicas, allowing the system to more effectively overcome energy barriers and ensuring a more thorough exploration of the conformational space.
- To ensure the statistical robustness and reproducibility of the observed unfolding pathways, it is standard practice to perform at least three independent replicates for each unique simulation condition (i.e., for each solvent-temperature pair). While the current study uses multiple temperatures, having multiple independent runs at each temperature would strengthen the conclusions by demonstrating that the observed pathways are not artifacts of a single stochastic trajectory.
- The native contacts defined as between two Cα While this is a common and computationally efficient approach, defining contacts based on all heavy atoms would provide a more detailed and chemically informative picture of the unfolding process. Heavy-atom contacts can better capture the disruption of side-chain packing and specific interactions.
- The use of Gaussian Mixture Model (GMM) clustering is a sophisticated choice for identifying conformational states. To ensure the reproducibility of this analysis, the Methods section should be expanded. It would be helpful to include specific details.
- In Figure 7, a structural comparison would more useful to show the difference.
- Minor grammatical and typographical errors should be corrected (e.g., “confrontation population” in Page 10 should be “conformational population”).
Author Response
Reply to Reviewer 1-Manuscript ID (biomolecules-3978498)
Reviewer 1
Comments and Suggestions for Authors
The manuscript presents a compelling molecular dynamics study on the unfolding pathways of the RRM domain of the Ewing Sarcoma (EWS) protein. The investigation of solvent-dependent conformational dynamics across a range of temperatures is well-motivated and addresses a significant gap in our understanding of this protein. The methodological approach is sound, and the combination of free energy landscapes, hydrogen bond analysis, and Gaussian mixture model clustering provides a robust framework for the conclusions.
I have several suggestions that I believe will further strengthen the manuscript.
Comment 1: Why the protein is solvent in 8M urea or 8M DMSO?
Response 1: We thank the reviewer for this insightful question. The protein was solvated in the aqueous solution of 8 M urea or 8 M DMSO to simulate the RRM of EWS in chemical denaturation conditions for investigating the unfolding mechanism underlying the different solvent. These high concentrations are widely used standard in both experimental and computational studies to induce complete unfolding of proteins environments [PMID: 12702764; 23536295; 24807152; 28803470; 15769860]. Thus, we opted the same.
Comment 2: The cumulative 5 µs of simulation time across multiple temperatures is substantial and significantly enhances conformational sampling. To further improve the efficiency and rigor of the sampling, the authors might consider employing Temperature Replica Exchange Molecular Dynamics (T-REMD). T-REMD facilitates the exchange of configurations between different temperature replicas, allowing the system to more effectively overcome energy barriers and ensuring a more thorough exploration of the conformational space.
Response 2: We sincerely thank the reviewer for this valuable suggestion. We fully acknowledge that Temperature Replica Exchange Molecular Dynamics (T-REMD) is a powerful enhanced-sampling technique that enables efficient crossing of energy barriers and improved conformational exploration through inter-replica temperature exchanges. Herein, we employed conventional MD simulations at in urea and DMSO, however, the temperature ranges are chosen as 300-500 K for both co-solvents relaying the notion that increasing temperature accelerates protein unfolding without changing the pathway of unfolding [PMID: 12215424; 12069627; 25296839]. This is the reason we able to observe complete unfolding during the time frame of 500 ns simulation at different temperatures in Urea as well as in DMSO, resulting to a cumulative sampling time of 5 µs to capture structural dynamics and different conformational transitions underlying the unfolding of HGT13. While T-REMD could indeed enhance sampling efficiency, it is computationally more demanding for membrane-protein systems of this size, as it requires running multiple replicas simultaneously. Nevertheless, we appreciate this insightful recommendation and agree that incorporating T-REMD in future work could provide an even more rigorous assessment of the folding–unfolding free energy landscape and conformational ensemble of HGT13 under diverse solvent and thermal conditions.
Comment 3: To ensure the statistical robustness and reproducibility of the observed unfolding pathways, it is standard practice to perform at least three independent replicates for each unique simulation condition (i.e., for each solvent-temperature pair). While the current study uses multiple temperatures, having multiple independent runs at each temperature would strengthen the conclusions by demonstrating that the observed pathways are not artifacts of a single stochastic trajectory.
Response 3: We sincerely thank the reviewer for this valuable and constructive suggestion. We fully agree that performing multiple independent replicates enhances the statistical robustness of MD simulations and ensures that the observed unfolding pathways are not artifacts of a single stochastic trajectory. In the present study, we carried out two independent set of simulations, in urea and DMSO at different temperatures, and these replicates consistently demonstrated similar unfolding trends and conformational transitions, supporting the reproducibility of our findings.
As per the reviewer’s recommendation, we have now additionally performed triplicate simulations, particularly at 400 K in both co-solvents, describing the conformational transition stabilities of different ensemble states, consistent with earlier results. These new simulations have been explicitly incorporated into the revised manuscript and figures updated in supplementary material as Figure S5. The updated results and discussion now reflect these replicates, providing further confidence in the robustness of the observed conformational transitions and stability of the ensemble states. Overall, the revised dataset offers an improved balance between computational feasibility and statistical reliability.
Comment 4: The native contacts defined as between two Cα While this is a common and computationally efficient approach, defining contacts based on all heavy atoms would provide a more detailed and chemically informative picture of the unfolding process. Heavy-atom contacts can better capture the disruption of side-chain packing and specific interactions.
Response 4: We thank the reviewer for this thoughtful suggestion. The method section has been updates as suggested.
Comment 5: The use of Gaussian Mixture Model (GMM) clustering is a sophisticated choice for identifying conformational states. To ensure the reproducibility of this analysis, the Methods section should be expanded. It would be helpful to include specific details.
Response 5: We thank the reviewer for appreciating our use of the Gaussian Mixture Model (GMM) approach and for the suggestion to improve reproducibility. The Methods section has now been expanded to include detailed information on the input features, training parameters, and validation procedures used in the GMM-based clustering. Specifically, the revised section now describes the inclusion of RMSD, Rg, fraction of native contacts (Q), number of hydrogen bonds, and secondary structure contents (from DSSP) as input descriptors. The GMM was trained using the Expectation–Maximization (EM) algorithm implemented in scikit-learn, and the Bayesian Information Criterion (BIC) was employed to determine the optimal number of clusters. The clustering results were further validated by assessing cluster stability across independent runs. All these details have been added to the Methods and Supplementary Materials (Figure S7) to ensure full transparency and reproducibility of the analysis.
Comment 6: In Figure 7, a structural comparison would more useful to show the difference.
Response 6: We thank the reviewer for this helpful suggestion. A comparative structural representation highlighting the differences among the representative conformations shown in Figure 7 has now been included in the Supplementary Materials. This addition provides a clearer visualization of the structural variations between the key conformational states and enhances the interpretability of Figure 7 in the revised manuscript.
Comment 7: Minor grammatical and typographical errors should be corrected (e.g., “confrontation population” in Page 10 should be “conformational population”).
Response 7: We thank the reviewer for carefully noting these errors. All grammatical and typographical mistakes, including the correction of “confrontation population” to “conformational population” (Page 10), have been thoroughly reviewed and corrected throughout the revised manuscript.